# Parallel encoding of sensory history and behavioral preference during *Caenorhabditis elegans* olfactory learning

Christine E Cho[1], Chantal Brueggemann[2], Noelle D L'Etoile[2], Cornelia I Bargmann[1]*

[1]Lulu and Anthony Wang Laboratory of Neural Circuits and Behavior, Howard Hughes Medical Institute, The Rockefeller University, New York, United States; [2]Departments of Cell and Tissue Biology and Medicine, University of California, San Francisco, United States

**Abstract** Sensory experience modifies behavior through both associative and non-associative learning. In *Caenorhabditis elegans,* pairing odor with food deprivation results in aversive olfactory learning, and pairing odor with food results in appetitive learning. Aversive learning requires nuclear translocation of the cGMP-dependent protein kinase EGL-4 in AWC olfactory neurons and an insulin signal from AIA interneurons. Here we show that the activity of neurons including AIA is acutely required during aversive, but not appetitive, learning. The AIA circuit and AGE-1, an insulin-regulated PI3 kinase, signal to AWC to drive nuclear enrichment of EGL-4 during conditioning. Odor exposure shifts the AWC dynamic range to higher odor concentrations regardless of food pairing or the AIA circuit, whereas AWC coupling to motor circuits is oppositely regulated by aversive and appetitive learning. These results suggest that non-associative sensory adaptation in AWC encodes odor history, while associative behavioral preference is encoded by altered AWC synaptic activity.

*For correspondence: cori@rockefeller.edu

**Competing interests:** The authors declare that no competing interests exist.

## Introduction

Sensory experience shapes sensory behavior. Primary sensory neurons adjust their sensitivity and dynamic range to capture ongoing sensory information without saturating, a phenomenon illustrated by the adaptation of the retina to ambient light levels over a $10^{10}$-fold range (*Arshavsky and Burns, 2012*; *Fain et al., 2001*). In addition, animals learn to increase, decrease, or switch their preference for sensory cues experienced in attractive or aversive contexts. Sensory adaptation and context-dependent learning affect overlapping circuits, which must preserve robust function in the face of continuously changing neuronal properties. Here, we show how an olfactory circuit implements these two processes by encoding sensory adaptation and aversive learning at distinct sites.

The nematode *Caenorhabditis elegans,* whose nervous system is composed of 302 neurons, shows robust plasticity in olfactory, mechanosensory, thermosensory, and gustatory behaviors (*Colbert et al., 1995*; *Rankin, 1991*; *Hedgecock and Russell, 1975*; *Saeki et al., 2001*). Olfaction may be its most complex sense. *C. elegans* detects hundreds of volatile odors using dedicated sensory neurons, each of which expresses multiple G protein-coupled receptors (GPCRs) (*Troemel et al., 1995*). For example, an olfactory neuron called AWC[ON] (one of two AWC neurons) detects benzaldehyde, butanone, and isoamyl alcohol, and expresses at least five chemosensory GPCRs (*Bargmann, 2006*; *Lesch and Bargmann, 2010*). Calcium imaging and genetic studies indicate that AWC signal transduction resembles mammalian phototransduction: odors are inferred to

**eLife digest** We learn from experience. When we repeatedly encounter a signal that is coupled to either reward or punishment, we eventually learn to expect the two to occur together. This phenomenon is called associative learning. Within the brain, distinct groups of neurons process information about the signal and about reward and punishment. In addition to storing information individually (as non-associative memories), the neurons communicate with one another and combine their information to create associations.

Like humans and many other animals, the roundworm and model organism *Caenorhabditis elegans* can learn to associate odors with rewards or punishments. By teaching worms that a scent predicts either food or a lack of food, Cho et al. now show that different cells and molecules support the formation of these two associations.

*C. elegans* detect odors using sensory neurons. Repeated exposure to an odor reduces a neuron's sensitivity to that odor, and Cho et al. show that this occurs irrespective of whether the odor is paired with reward or with punishment. This indicates that the neuron stores information about the odor as a non-associative memory. By contrast, pairing an odor with reward has differing effects on associative learning to pairing that same odor with punishment. Pairing an odor with a reward increases a sensory neuron's ability to communicate with target neurons – ultimately, those that control movement – whereas odor-punishment pairing reduces this ability. Further experiments showed that an insulin peptide supports learning about odors and punishments, but not about odors and rewards.

The next challenge is to identify the molecules that strengthen or weaken communication between sensory neurons and target neurons after associative learning. It will also be important to identify the other neurons and molecules that detect rewards and punishments, to gain a more complete picture of how the brain acquires this information.

decrease the level of cGMP, close a cGMP-dependent transduction channel, and hyperpolarize AWC (*Bargmann, 2006*; *Chalasani et al., 2007*).

When *C. elegans* is exposed to high concentrations of an odor in the absence of food, it gradually loses its attraction to that odor over an hour or more, and recovers over a similar timescale (*Colbert and Bargmann, 1995*). This process has been called adaptation, but will here be called aversive olfactory learning to reflect its long duration and the required pairing with food deprivation (*Nuttley et al., 2002*), and to distinguish it from short-term sensory adaptation. Aversive learning is selective for the experienced odor, even when two odors are sensed by partly overlapping olfactory neurons (*Colbert and Bargmann, 1995*). The genetic requirements for aversive learning vary depending on the odor and the duration of odor exposure, and include G protein signaling pathways, ion channels, and transcriptional regulators (*Ardiel and Rankin, 2010*).

The cGMP-dependent protein kinase EGL-4 is closely associated with aversive learning. *egl-4* mutants show learning defects to multiple AWC-sensed odors, among other sensory defects (*Daniels et al., 2000*; *L'Etoile et al., 2002*). After prolonged odor conditioning, EGL-4 translocates from the AWC cytoplasm to the nucleus (*O'Halloran et al., 2009*; *Lee et al., 2010*), where it phosphorylates the heterochromatin protein HPL-2 and alters gene expression (*Juang et al., 2013*). Nuclear translocation of EGL-4 is a real-time marker for AWC plasticity (*Lee et al., 2010*). EGL-4 translocation requires olfactory signal transduction and endogenous cGMP signaling in AWC, but exogenous cGMP is not sufficient to modify EGL-4 localization (*O'Halloran et al., 2012*), suggesting that a second coincident signal is required.

The existence of a coincident context signal for learning is further supported by the observation that aversive learning only occurs when odors are paired with food deprivation (*Nuttley et al., 2002*). Indeed, butanone odor becomes more attractive after pairing with food, indicating that food context drives bidirectional olfactory learning (*Torayama et al., 2007*; *Kauffmann et al., 2010*). The integration of food context may involve communication between neurons, as aversive learning requires insulin signals from other neurons that feed back onto AWC and reduce its activity (*Chalasani et al., 2010*; *Lin et al., 2010*).

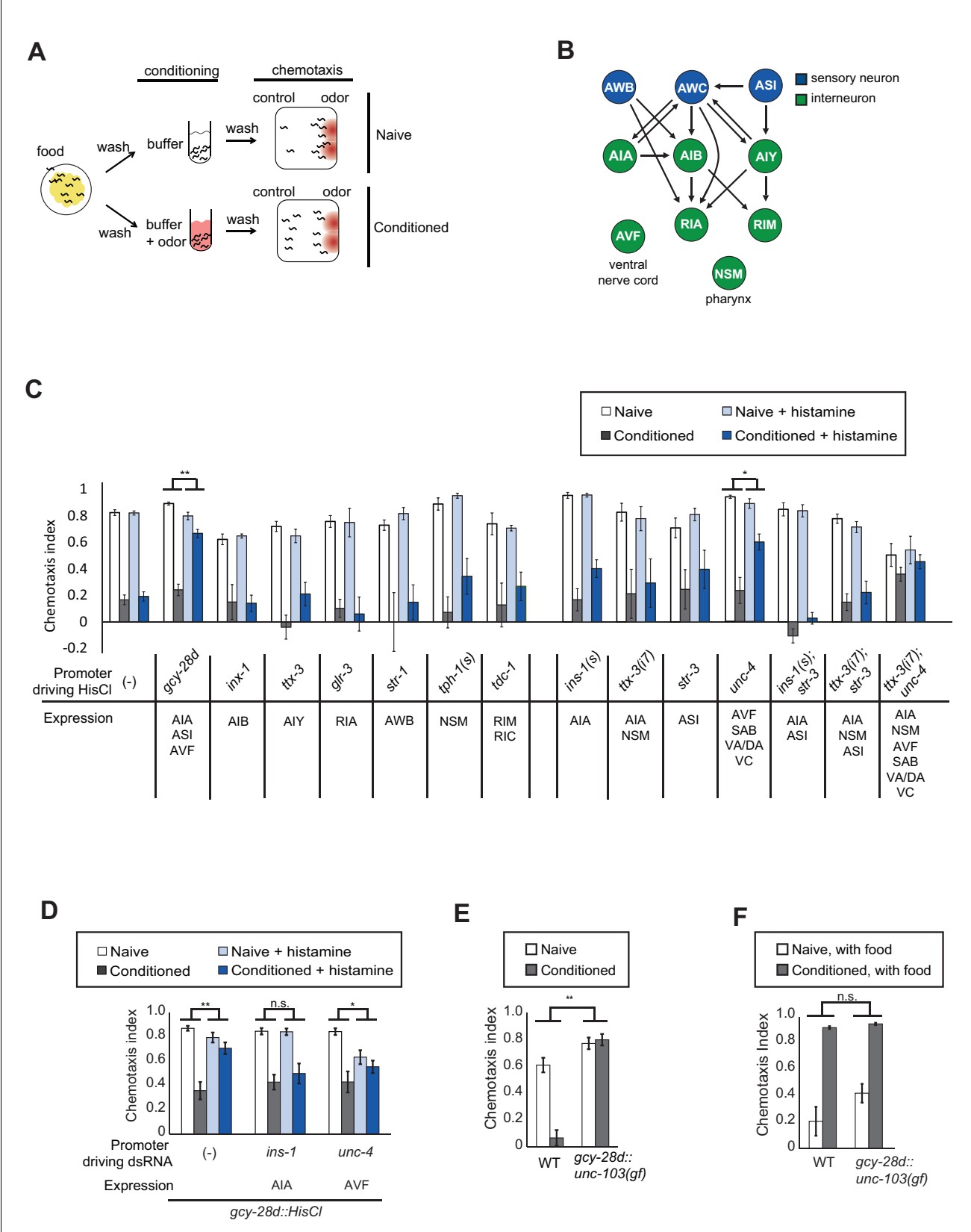

**Figure 1.** Neurons required during odor conditioning for aversive olfactory learning. (A) Schematic of aversive olfactory learning assay. Adult worms are washed off food and conditioned with odor in buffer for 90 min, then washed again before being tested in a butanone chemotaxis assay for 1–2 hr. (B) *Figure 1 continued on next page*

*Figure 1 continued*

Partial *C. elegans* wiring diagram showing neurons tested for effects on aversive learning and their synaptic connections with AWC and each other. (**C**) Aversive learning in animals expressing the histamine-gated chloride channel (HisCl1) under cell-specific promoters, assayed with or without 10 mM histamine in the conditioning medium. Chemotaxis assays (1:1000 butanone dilution) were performed on histamine-free plates. Error bars represent S.E. M. *P* values were generated by 2-way ANOVA for interaction of odor condition and presence of histamine (**p<0.001, *p<0.05). n = 3–61 assays, 50–200 animals/assay. (**D**) Aversive learning in animals carrying the *gcy-28d::HisCl* transgene and a second transgene expressing dsRNA that targets HisCl. Tested as in (**B**). Error bars represent S.E.M. *P* values were generated by 2-way ANOVA for interaction of odor condition and presence of histamine (**p<0.001, *p<0.05, *n.s.* not significant). n = 9–15 assays, 50–200 animals/assay. (**E**) Aversive learning in animals expressing the gain-of-function potassium channel UNC-103 under the *gcy-28d* promoter. *P* values were generated by 2-way ANOVA for interaction of genotype and condition (**p<0.001). n = 8 assays, 50–200 animals/assay. (**F**) Appetitive learning in wild-type and *gcy-28d::unc-103(gf)* animals after conditioning with butanone and food (1:10 butanone dilution). 2-way ANOVA for interaction of genotype and condition (*n.s.* not significant). n = 8 assays per condition, 50–200 animals/assay.

The following source data is available for figure 1:

**Source data 1.** Individual chemotaxis indices for *Figure 1C–F*.

The circuits for aversive olfactory learning are only partially understood. First, it is unclear how information about odor and food context is represented and associated during odor conditioning. Second, the neuronal sites at which information is stored are unknown. Here, we use circuit manipulations, molecular markers, and in vivo calcium imaging to map the effects of odor history and food context on the olfactory circuit. We show that during aversive learning, the food-deprivation context engages feedback from AIA and other neurons, as well as insulin signaling. This feedback regulates EGL-4 localization and the behavioral output of AWC. In parallel, non-associative sensory adaptation shifts the dynamic range of AWC during both appetitive and aversive learning. Thus olfactory learning induces both associative and non-associative plasticity in a single sensory neuron.

## Results

### AIA and other neurons are required for aversive olfactory learning

*C. elegans* is strongly attracted to butanone, an odor sensed by the AWC$^{ON}$ olfactory neuron, but this attraction is lost after butanone is paired with food deprivation for 90 min (*Figure 1A*). To identify neurons required for olfactory learning, we acutely silenced neurons that form pre- and postsynaptic connections with AWC in the *C. elegans* wiring diagram (*Figure 1B*; *White et al., 1986*). Small groups of neurons were targeted by cell-specific expression of the *Drosophila* histamine-gated chloride channel HisCl1, and silenced by administration of exogenous histamine during the conditioning period (*Pokala et al., 2014*). This temporary inactivation should identify neurons that function in learning, rather than neurons with general effects on locomotor behaviors or chemotaxis strategies.

Among a panel of tested strains, aversive learning was nearly eliminated in *gcy-28d::HisCl1* animals that were exposed to histamine during conditioning (*Figure 1C*). This transgene is expressed reliably in AIA (100%) and AVF (90%) neurons and less frequently in ASI, I1, IL2 and M3 neurons (see Materials and methods); AIA and ASI form synapses with AWC. No defect was observed after HisCl1 silencing of several other AWC synaptic partners (AIB, AIY, RIA), other sensory neurons (AWB), or neuromodulatory neurons (NSM, RIM, RIC). Control experiments demonstrated that histamine did not affect butanone chemotaxis or aversive learning in wild-type animals, and that the HisCl1 transgenes were innocuous in the absence of histamine (*Figure 1C*). Thus *gcy-28d::HisCl1* selectively silences neurons required during the conditioning period for aversive olfactory learning.

To narrow down the relevant neurons expressing *gcy-28d::HisCl1*, we took two complementary approaches: expressing HisCl1 in subsets of the *gcy-28d*-expressing neurons in a wild-type strain, and expressing double-stranded HisCl1 RNA in subsets of neurons to silence transgene expression in the *gcy-28d::HisCl1* strain. We were unable to identify a single neuron in which HisCl1 expression could replicate the strong learning defect of the *gcy-28d::HisCl1* strain (*Figure 1C*). The most promising initial candidate in the *gcy-28d* set was AVF, because an *unc-4::HisCl1* transgene that is expressed in AVF caused a partial learning defect (*Figure 1C*). However, *unc-4::HisCl1* is expressed in many neurons other than AVF that might contribute to its effect. Moreover, dsRNA-induced

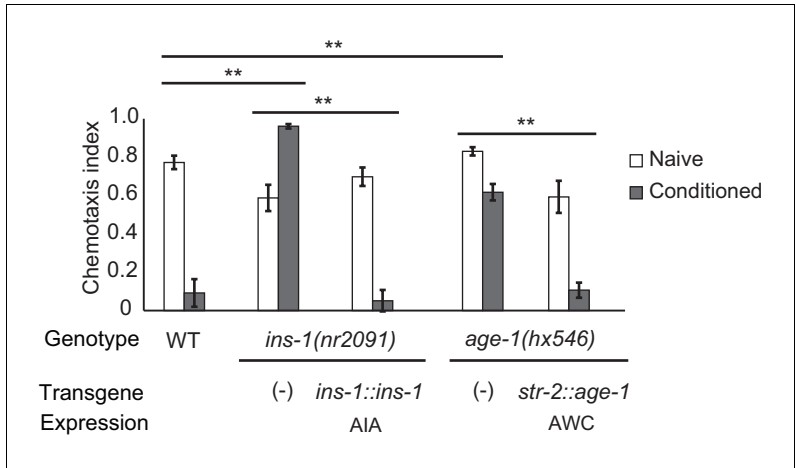

**Figure 2.** Cell-specific requirements for the insulin signaling pathway in aversive olfactory learning. Aversive olfactory learning in mutants of the insulin signaling pathway, with or without cell-specific transgenes expressing cDNAs for the insulin gene *ins-1* and the PI3 kinase gene *age-1*. Error bars represent S.E.M. *P* values were generated by 2-way ANOVA for interaction of genotype and condition (**All comparisons significant at p<0.01 after Bonferroni correction). n = 4–26 assays, 50–200 animals/assay.

The following source data is available for figure 2:

**Source data 1.** Individual chemotaxis indices for *Figure 2*.

silencing of the *gcy-28d::HisCl1* transgene in AVF did not rescue aversive learning, but instead caused a mild chemotaxis defect (*Figure 1D*). These results suggest that AVF might affect chemotaxis rather than learning per se.

Silencing the *gcy-28d::HisCl1* transgene in AIA consistently restored aversive learning (*Figure 1D*), a result that implicates AIA in learning. However, silencing AIA alone or in several combinations with other neurons with HisCl1 did not disrupt learning (*Figure 1C*). Combining transgenes that express HisCl in AIA and AVF resulted in a learning defect even in the absence of histamine, an ambiguous result. These results suggest that simultaneous silencing of AIA and at least one unidentified neuron cause the aversive learning defect in the *gcy-28d::HisCl1* strain.

As an alternative chronic silencing method, we expressed an overactive UNC-103 potassium channel under the *gcy-28d* promoter. This transgene resulted in a strong defect in aversive olfactory learning (*Figure 1E*). The *gcy-28d::unc-103(gf)* transgene is expressed reliably in AIA interneurons and only occasionally in other neurons (see Materials and methods), supporting the hypothesis that AIA promotes aversive learning. For simplicity, the neurons affected by this *gcy-28d::unc-103(gf)* transgene will be described as 'the AIA circuit,' with the recognition that other neurons may also contribute to its effects.

Pairing butanone with food results in increased attraction to butanone, a phenomenon called appetitive olfactory learning or butanone enhancement (*Torayama et al., 2007*; *Kauffman et al., 2010*). Appetitive olfactory learning was unaffected by silencing the AIA circuit with the *gcy-28d::unc-103(gf)* transgene (*Figure 1F*), indicating a preferential requirement for the AIA circuit in aversive learning.

## Insulin and PI3 kinase signaling promote aversive learning

The AIA interneurons receive synaptic input from numerous chemosensory neurons, and make reciprocal synapses onto a few, including AWC. Previous studies have implicated retrograde insulin (*ins-1*) signaling from AIA to AWC in aversive learning to odors including benzaldehyde and isoamyl alcohol (*Lin et al., 2010*; *Chalasani et al., 2010*). We found that aversive learning to butanone was also lost in the *ins-1* mutant, and was rescued by expressing *ins-1* in the AIA neurons (*Figure 2*).

INS-1 and other insulin-like proteins act on the insulin receptor DAF-2, which regulates the AGE-1 phosphatidyl inositol-3-kinase (PI3K) and downstream kinases and transcription factors (*Murphy and*

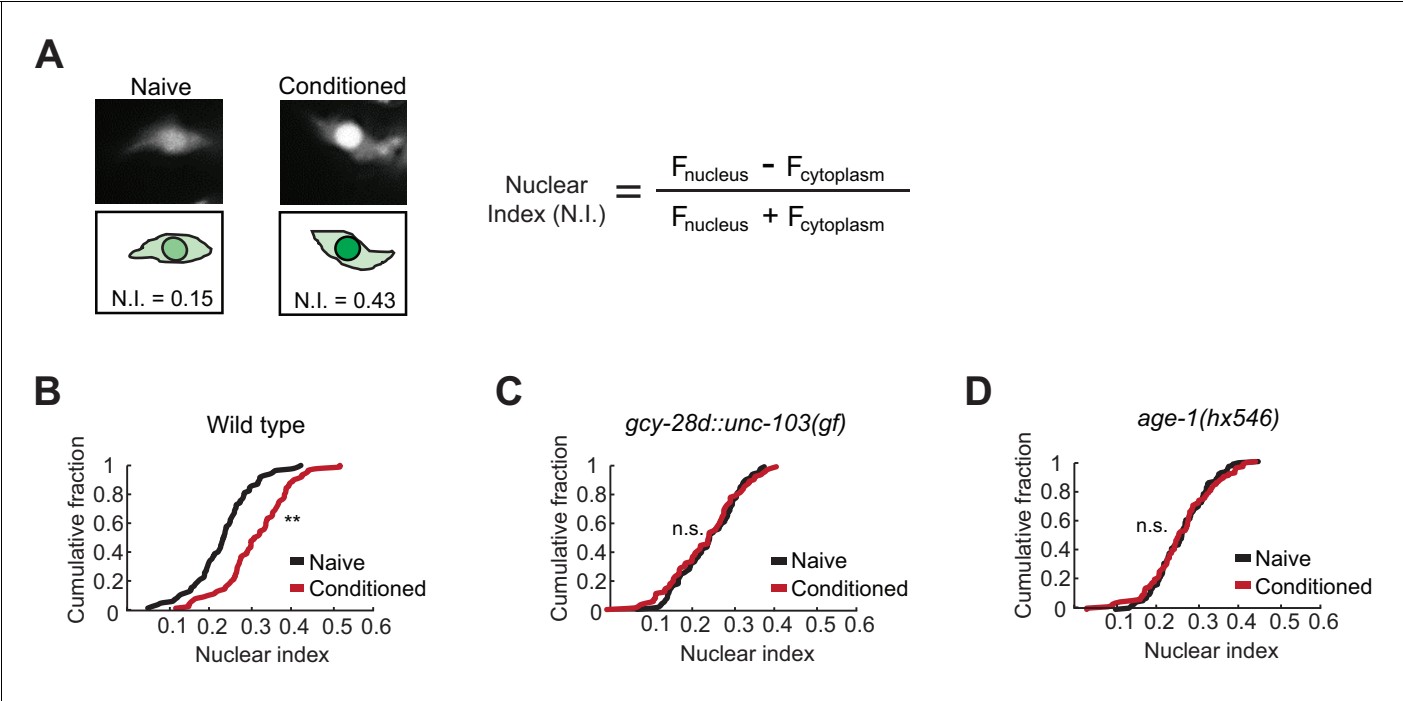

**Figure 3.** Nuclear enrichment of EGL-4 in AWC neurons after odor conditioning. (**A**) Representative images of EGL-4::GFP fluorescence and nuclear index in the AWC of naive and conditioned animals (left), and the equation used to quantify the degree of EGL-4 nuclear localization (right). $F_{nucleus}$, $F_{cytoplasm}$ = fluorescence measured in AWC nucleus or cytoplasm of the same neuron. (**B,C,D**) Cumulative distribution (AWC nuclear index) for EGL-4:: GFP in wild-type, *gcy-28d::unc-103(gf)*, and *age-1(hx546)* animals after conditioning. An increase in AWC nuclear index is observed after conditioning wild-type (**B**) but not *gcy-28d::unc-103(gf)* (**C**) or *age-1(hx546)* (**D**). *P* values were generated by nonparametric Kolmogorov-Smirnov test (**p<0.001). n = 79–90 animals per condition.

The following source data is available for figure 3:

**Source data 1.** Individual nuclear indices for *Figure 3B,C,D*.

---

*Hu, 2013*). *daf-2* and *age-1* null mutants are inviable, but viable *age-1* reduction of function mutants were defective in aversive olfactory learning, and could be rescued by expressing *age-1* in AWC[ON] (*Figure 2*). Thus insulin signaling to AWC is one possible mechanism by which the AIA circuit could regulate aversive learning to butanone.

## The AIA circuit drives nuclear enrichment of EGL-4 in AWC during learning

To understand how the AIA circuit modifies olfactory behavior, we examined the effect of AIA silencing on reporters of AWC activity and olfactory plasticity. We began with the cGMP-dependent protein kinase EGL-4, whose translocation from the cytoplasm to the nucleus of AWC neurons during odor conditioning is essential for aversive olfactory learning (*Lee et al., 2010*).

Quantitative microscopic analysis of an EGL-4::GFP transgene expressed in AWC neurons was used to determine cytoplasmic and nuclear levels of EGL-4 after pairing odor with food deprivation (*Figure 3A*). In agreement with previous results, odor conditioning led to enrichment of EGL-4::GFP in the AWC nucleus within 90 min (*Figure 3B*). However, when the AIA circuit was silenced, naive and butanone-conditioned animals had indistinguishable EGL-4::GFP localization in AWC, resembling naive wild-type animals (*Figure 3C*). These results identify nuclear enrichment of EGL-4::GFP as a cell-biological readout of signaling from the AIA circuit to AWC.

The *ins-1/age-1* signal from AIA to AWC is a candidate to mediate the effect of the AIA circuit on AWC EGL-4::GFP during learning. Indeed, *age-1* (PI3K) reduction of function mutants, which were defective in aversive learning (*Figure 2*), also failed to relocalize EGL-4::GFP in AWC after odor conditioning (*Figure 3D*).

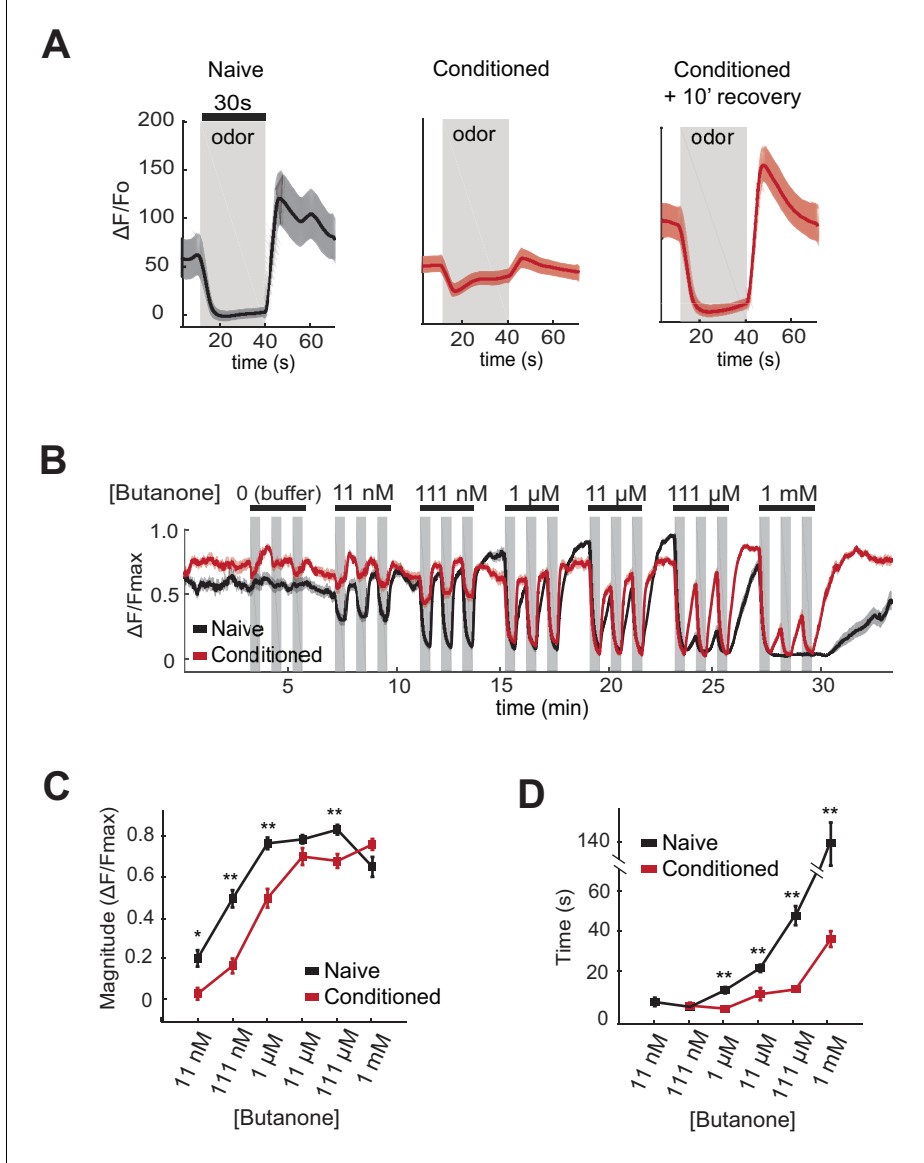

**Figure 4.** AWC[ON] butanone responses shift after odor conditioning. (**A**) Average AWC[ON] calcium responses to a 30-second pulse of 1 μM butanone in naive animals, conditioned animals, or conditioned animals after 10 min of recovery in buffer. Gray represents odor. Shaded region represents S.E.M. n = 8–27 animals. (**B**) AWC[ON] calcium responses of naive animals and conditioned animals to a range of butanone concentrations. Animals were washed in buffer for 15 min after conditioning. Gray represents odor (30 s pulses). Shaded region represents S.E.M. n = 25–26 animals. (**C**) Average response magnitude after butanone addition (first of three pulses, data from (**B**)). (**D**) Half-time of recovery after butanone removal (last of three pulses, data from (**B**)). P values were generated by t-test at each odor concentration with correction for unequal variance (**p<0.001, *p<0.05). Error bars represent S.E.M.

The following source data is available for figure 4:

**Source data 1.** Data and heat map showing individual responses for *Figure 4B*.
**Source data 2.** Data for response magnitude and recovery time in *Figure 4C,D*.

## Odor conditioning shifts the dynamic range of butanone responses in AWC<sup>ON</sup>

One potential mechanism by which aversive learning could suppress butanone attraction is downregulation of olfactory signal transduction in AWC<sup>ON</sup>. Attractive odors reduce AWC<sup>ON</sup> calcium, and subsequent odor removal results in a calcium increase that can overshoot before returning to baseline (*Chalasani et al., 2007*). Previous studies have demonstrated a near-complete loss of AWC odor sensitivity after an hour of conditioning with isoamyl alcohol (*Chalasani et al., 2010*). Using genetically-encoded calcium indicators to monitor butanone responses after aversive learning, we found that AWC<sup>ON</sup> calcium responses at 1 µM butanone were greatly diminished after butanone conditioning (*Figure 4A*). However, the AWC<sup>ON</sup> responses were restored after ten minutes of recovery in buffer (*Figure 4A*). This short-term suppression of butanone responses appears insufficient to explain aversive learning behavior, which persists through an hour-long chemotaxis assay (*Figure 1A*).

To obtain a more quantitative understanding of sensory dynamics, AWC<sup>ON</sup> calcium responses were examined in a high-throughput system that allowed simultaneous calcium imaging in multiple animals across multiple odor pulses (*Larsch et al., 2013*). A dose-response curve of naive animals showed that AWC<sup>ON</sup> calcium is suppressed by butanone over a $10^5$-fold concentration range from 11 nM to 1 mM. This suppression of basal calcium by odor saturated between 111 nM and 1 µM butanone, was followed by a significant calcium overshoot after odor removal at 11 nM to 11 µM, and recovered to baseline only slowly from odor concentrations above 11 µM (*Figure 4B–D*, black traces).

After pairing butanone conditioning with food deprivation, the dynamic range of AWC<sup>ON</sup> calcium responses was shifted toward higher concentrations (*Figure 4B,C*; red traces). Butanone-conditioned animals had a ten-fold increase in the detection threshold, a ten-fold increase in the saturation concentration, a faster recovery after odor removal, and a reduced calcium overshoot compared to control animals conditioned in buffer alone (*Figure 4B–D*). The shift in dynamic range persisted for at least 40 min, consistent with the time course of aversive learning.

## Changes in AWC<sup>ON</sup> dynamic range reflect odor history, not odor preference

The effects of the AIA circuit on AWC<sup>ON</sup> dynamic range were examined by calcium imaging in the *gcy-28d::unc-103(gf)* strain. Naive AWC<sup>ON</sup> calcium responses in this strain resembled those of wild-type animals (*Figure 5A*, black traces). Surprisingly, butanone conditioning resulted in the same shift in the AWC<sup>ON</sup> threshold, saturation concentration, and recovery dynamics as in the wild type (*Figure 5A–C*, red traces), despite the fact that these animals did not show aversive learning at a behavioral level. This result demonstrates that the shift in AWC<sup>ON</sup> dynamic range after conditioning is not sufficient for learning.

The AWC<sup>ON</sup> dynamic range also shifted after butanone conditioning in *age-1* PI3K mutants, which have a learning defect (*Figure 5D–F*). In addition, conditioned *age-1* mutants had pronounced AWC<sup>ON</sup> oscillations after odor removal, as previously reported for *ins-1* mutants (*Figure 5—Source data 1*; *Chalasani et al., 2010*). *age-1* probably affects several signaling processes, but its response supports the conclusion that a shift in AWC<sup>ON</sup> dynamic range is insufficient for aversive learning.

Finally, we monitored odor responses in AWC<sup>ON</sup> neurons after appetitive learning. Behavioral attraction to butanone is enhanced after butanone is paired with food (*Figure 1F*), but AWC<sup>ON</sup> was less sensitive to butanone in these conditioned animals than in the matched naive control (*Figure 5G–I*). Indeed, the AWC<sup>ON</sup> calcium response was similarly affected by appetitive and aversive conditioning, with a ten-fold increase in detection threshold, an increase in the saturation concentration, and a faster recovery after odor removal compared to naive controls (*Figure 5H,I*). Thus the shift in butanone dynamic range in AWC<sup>ON</sup> represents sensory adaptation to odor history, and not odor preference.

This experiment also revealed that food deprivation alters AWC<sup>ON</sup> responses: food-deprived naive animals (*Figure 4B,C*) detected lower butanone concentrations than well-fed naive animals (*Figure 5G,H*). The calcium-calmodulin dependent protein kinase CMK-1, which shuttles between the AWC nucleus and cytoplasm in response to food deprivation, is a candidate to mediate this change in odor sensitivity (*Neal et al., 2015*).

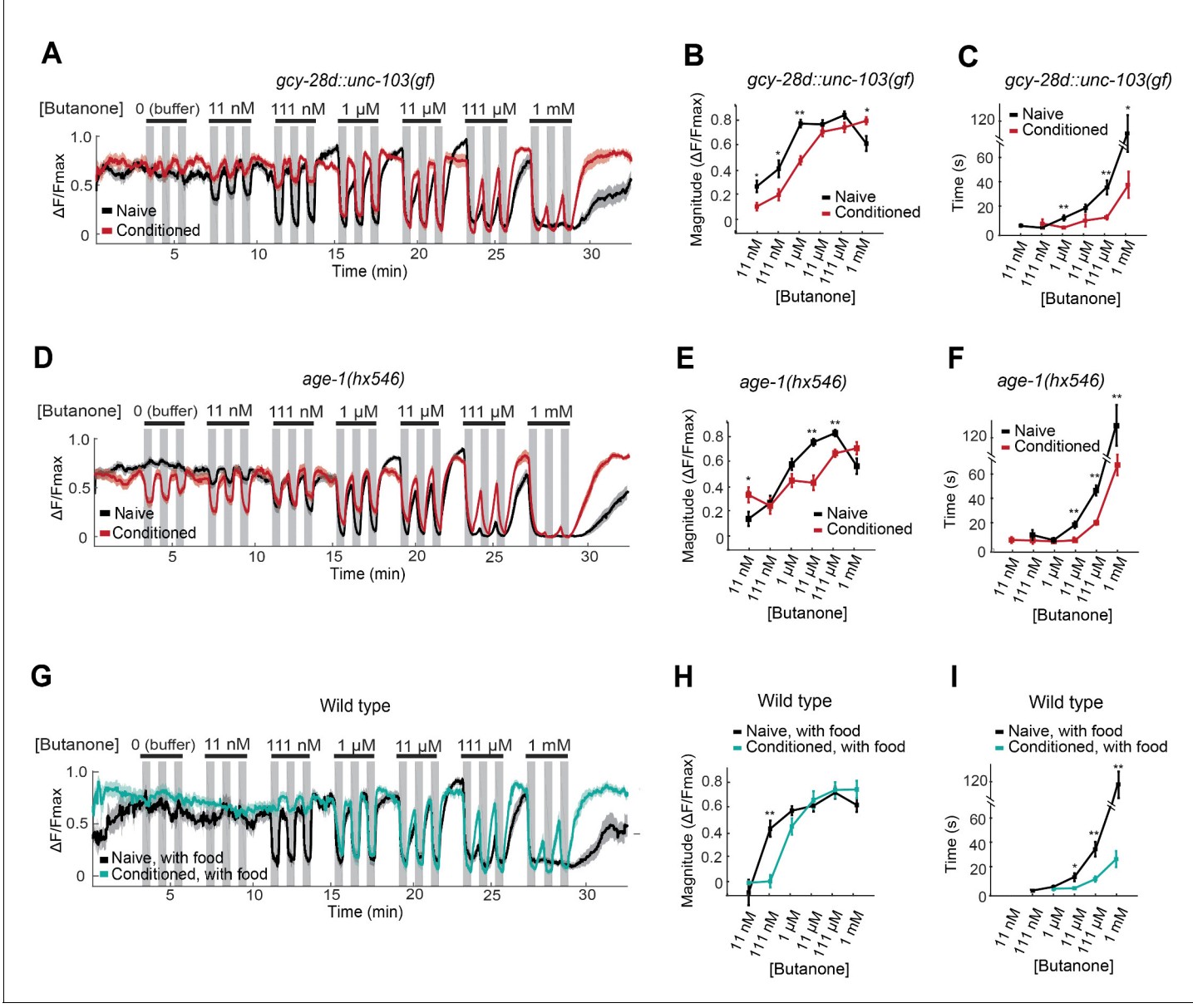

**Figure 5.** AWC[ON] butanone responses report odor history. (**A–F**) AWC[ON] calcium responses are altered by aversive conditioning in mutants that do not learn. (**A–C**) *gcy-28d::unc-103(gf)* (**D–F**) *age-1(hx546)*. (**G–I**) AWC[ON] calcium responses of wild-type animals after appetitive conditioning with odor and food. Note that the well-fed naive and conditioned groups in (**G–I**) are less sensitive to odor than the food-deprived groups in *Figure 4*. (**A,D,G**) Calcium responses to a concentration series with three 30 sec pulses per concentration. Gray represents odor. (**B,E,H**) Average response magnitude after butanone addition (first of three pulses). (**C,F,I**) Half-time of recovery after butanone removal (last pulse). n = 11–24 animals. *P* values were generated by t-test at each odor concentration with correction for unequal variance (**p<0.001, *p<0.05). Error bars in (**B,C,E,F,H,I**) and shaded regions in (**A,D,G**) represent S.E.M.

The following source data is available for figure 5:

**Source data 1.** Data and heat map showing individual responses in *Figure 5A,D,G*.
**Source data 2.** Data for response magnitude and recovery time in *Figure 5B,C,E,F,H,I*.

## AWC-induced reversal behavior reflects bidirectional odor preference

The AWC neurons can drive acute behavioral responses as well as long-range chemotaxis, providing an additional assay for AWC function before and after learning. *C. elegans* locomotion alternates

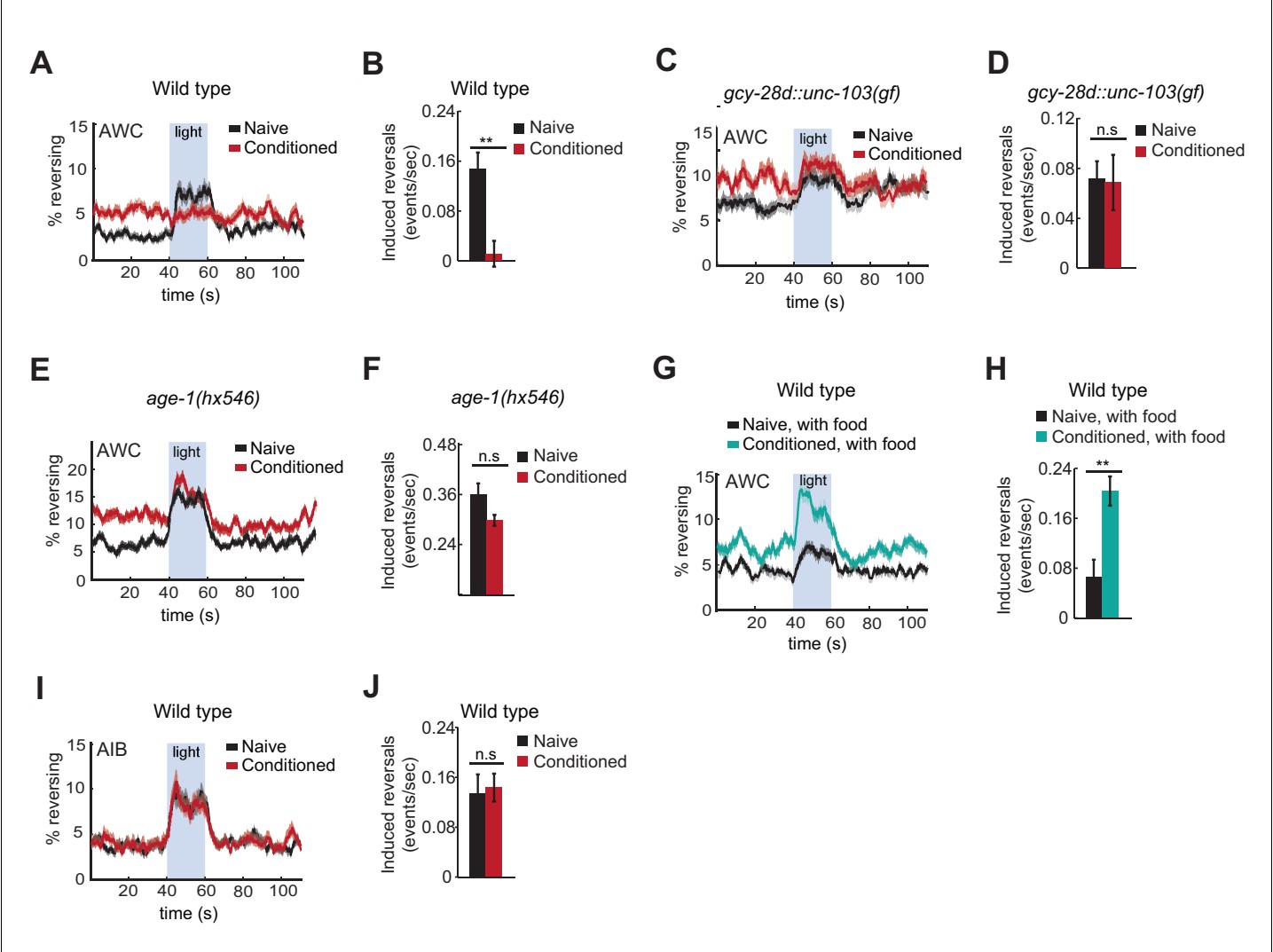

**Figure 6.** Behavioral responses to acute neuronal activation after odor conditioning. (A,B) Light-induced reversals in naive or conditioned wild-type animals expressing Channelrhodopsin (ChR2) in AWC[ON]. Data show the average fraction of animals executing reversals over time (A), or the difference between the number of reversals initiated during and after stimulation (20 sec each, B). (C,D) Light-induced reversals in naive or conditioned *gcy-28d:: unc-103(gf)* animals expressing AWC[ON]::ChR2. Note increased basal frequency of reversals, a known property of AIA inactivation (*Chalasani et al., 2010*). (E,F) Light-induced reversals in naive or conditioned *age-1(hx546)* animals expressing AWC[ON]::ChR2. (G,H) Light-induced reversals in wild-type AWC[ON]::ChR2 animals after appetitive conditioning. (I,J) Light-induced reversals in AIB::ChR2 animals after aversive conditioning. Pale blue regions in (A,C,E,G,I) represent blue light stimulation. Shaded regions and error bars represents S.E.M. n = 7–14 assays per condition, 18–25 animals stimulated five times per assay. *P* values were generated by t-test with correction for unequal variance (**p<0.001, *n.s.* not significant).

The following source data is available for figure 6:

**Source data 1.** Data for induced reversal frequency in *Figure 6B,D,F,H,J*.

between forward runs and reversals, which are regulated by odors during chemotaxis (*Pierce-Shimomura et al., 1999*; *Iino and Yoshida, 2009*). Odor removal increases AWC[ON] activity and the frequency of reversals in wild-type animals, but not in chemotaxis-defective mutants, suggesting that the acute reversal assay and chemotaxis are related (*Albrecht and Bargmann, 2011*). Like odor removal, direct activation of AWC[ON] with the light-activated ion channel Channelrhodopsin 2 (ChR2) elicits reversals (*Gordus et al., 2015*). AWC[ON]::ChR2 bypasses odor transduction to depolarize AWC directly, thereby separating AWC sensory processing from AWC coupling to behavioral circuits.

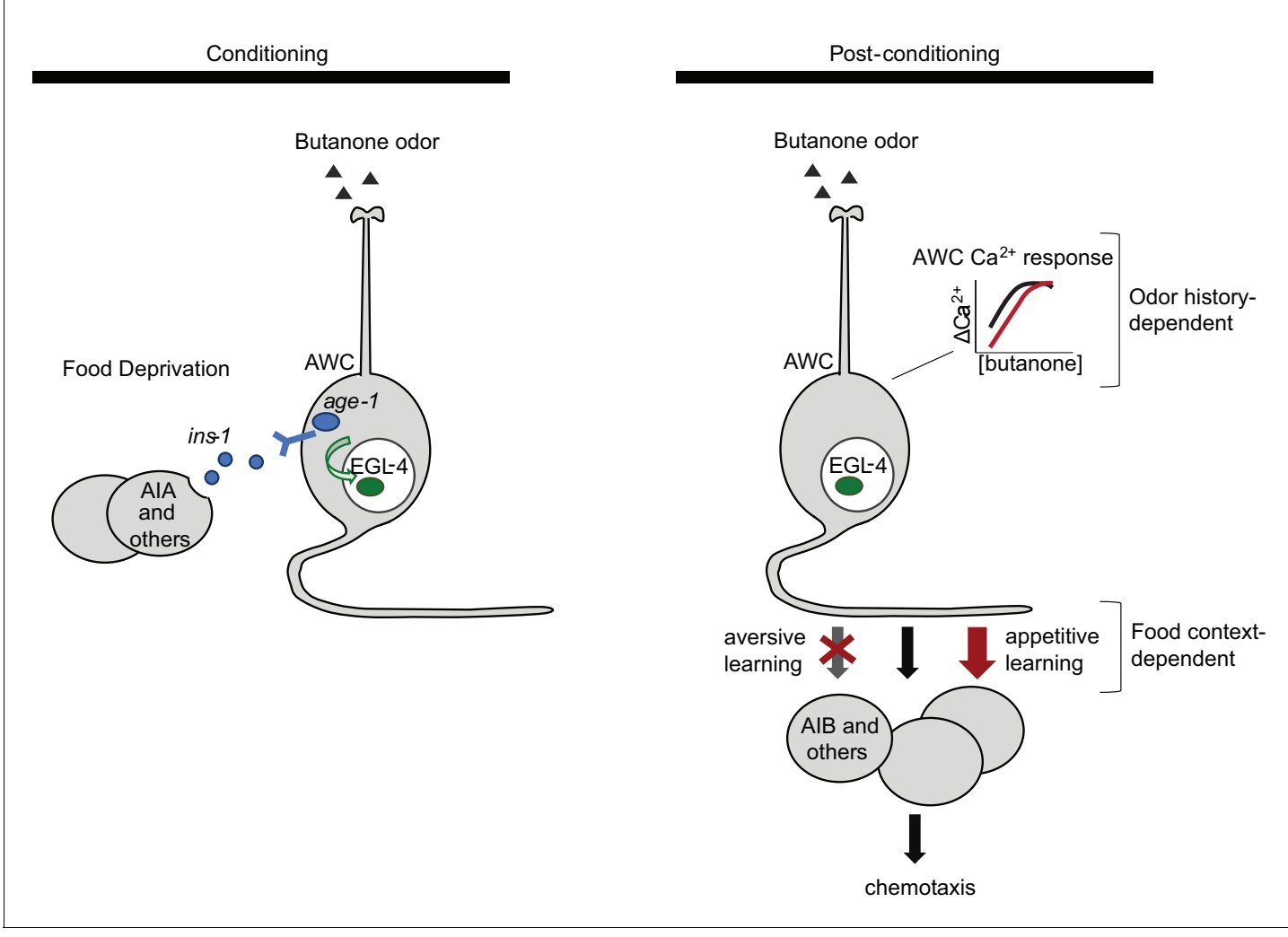

**Figure 7.** Two sites of plasticity in aversive olfactory learning. *Left* Odor conditioning. During pairing of odor with food deprivation, retrograde signals from the AIA circuit to AWC^ON lead to EGL-4 accumulation in the AWC nucleus. This molecular event is associated with changes in AWC^ON gene expression (*Juang et al., 2013*), and is required for aversive olfactory learning (*Lee et al., 2010*). *Right* Odor experience has two distinct effects on AWC^ON. A similar shift in AWC^ON dynamic range occurs after both aversive and appetitive conditioning, representing non-associative sensory adaptation. A change in AWC^ON synaptic output is associative and bidirectional, leading to opposite behavioral changes in aversive and appetitive learning. The AIA circuit is required only for the change in AWC synaptic output.

To ask how learning affects AWC behavioral output, AWC^ON::ChR2-expressing animals were conditioned with butanone and food deprivation and then stimulated with blue light while their locomotion was recorded. Naive animals responded to AWC^ON activation with an increase in reversals, as previously reported (*Figure 6A,B*). By contrast, butanone-conditioned animals did not reverse in response to AWC^ON::ChR2 stimulation (*Figure 6A,B*). This result suggests that aversive conditioning depresses AWC^ON coupling to target neurons that drive reversal behaviors.

Suppression of the AWC^ON::ChR2 response required the AIA circuit, as *gcy-28d::unc-103(gf)* animals responded equally strongly to AWC^ON::ChR2 stimulation with or without butanone conditioning (*Figure 6C,D*). Similarly, the behavioral response to AWC^ON::ChR2 stimulation in *age-1* mutants was not significantly altered by conditioning, matching their failure in aversive learning (*Figure 6E, F*).

Finally, we tested the effects of appetitive conditioning on AWC output using the same assay. After butanone was paired with food, AWC^ON::ChR2 stimulation elicited a substantially greater increase in reversals than it did in naive controls (*Figure 6G,H*). Therefore, AWC^ON::ChR2-driven

behavioral output demonstrates bidirectional plasticity after conditioning that mirrors aversive and appetitive learning behaviors.

These results indicate that odor conditioning alters the coupling of AWC to motor circuits. To narrow down the locus at which this change might occur, we examined AIB, an interneuron that receives direct and indirect synaptic output from AWC. Channelrhodopsin activation of AIB elicits reversals in naive animals, like activation of AWC (*Gordus et al., 2015*; *Piggott et al., 2011*). AIB::ChR2 illumination elicited reversals equally well in naive and butanone-conditioned animals, localizing plasticity to a site upstream of or parallel to AIB (*Figure 6I,J*).

## Discussion

Butanone conditioning modifies two aspects of AWC$^{ON}$ function (*Figure 7*). First, the dynamic range of AWC$^{ON}$ calcium signaling shifts toward higher butanone concentrations after conditioning. This alteration in AWC$^{ON}$ is equivalently induced by aversive and appetitive conditioning, indicating that AWC sensitivity is a non-associative representation of odor history. Second, the coupling of AWC$^{ON}$ activity with reversal behavior decreases after odor is paired with food deprivation, and increases after odor is paired with food. This bidirectional plasticity is associative; it integrates odor experience with food context and mirrors the behavioral preference after learning.

Silencing AIA and other *gcy-28d*-expressing neurons acutely during conditioning prevents aversive learning. In addition to AIA, the relevant neurons might include ASI, which expresses *ins-1* and has redundant roles with AIA in aversive learning to benzaldehyde (*Lin et al., 2010*), and AVF, a little-characterized neuron with a role in locomotion (*Hardaker et al., 2001*). The effects of the AIA circuit on aversive learning are similar to those of the insulin-like peptide *ins-1*, which is expressed in AIA, ASI, and other neurons. As well as having similar behavioral effects, the AIA circuit and insulin/ PI3 kinase signaling have similar effects on nuclear translocation of EGL-4 in AWC$^{ON}$, which is essential for aversive learning (*Lee et al., 2010*). INS-1 and downstream PI3 kinase signaling have previously been implicated in retrograde signaling from AIA to sensory neurons in several behavioral paradigms including salt chemotaxis learning and benzaldehyde associative plasticity (*Tomioka et al., 2006*; *Lin et al., 2010*; *Chalasani et al., 2010*). It will be interesting to ask whether these also engage EGL-4 translocation as a signaling mechanism.

INS-1 is one of over 40 insulin-like peptides that may be agonists or antagonists of the insulin receptor DAF-2 (*Murphy and Hu 2013*). Given this diversity, INS-1 may not be unique in its ability to regulate aversive learning; under other circumstances, the PI3 kinase pathway in AWC could respond to other insulins secreted from other cells.

### Aversive learning has analogies with classical conditioning

Studies of associative learning in many animals have delineated sensory pathways that represent the conditioned stimulus (CS), the unconditioned stimulus (US), and their convergence point within the neural circuit. In the best-known examples, sites of convergence are in higher brain areas. In *Drosophila* odor learning, olfactory CS information converges with electric shock or sucrose reward US information in the mushroom bodies (*Waddell, 2010*). In rodents, auditory CS information converges with electric shock US information in the amygdala (*Romanski et al., 1993*). However, there are also examples of convergence in early sensory areas. In *Aplysia*, classical conditioning pairs an electric shock US with a gentle tactile CS to strengthen synapses of the tactile CS sensory neurons (*Hawkins et al., 1983*). In the visually-mediated ciliary response of *Hermissenda*, CS and US pathways converge at two sites – first-order interneurons shared by the two pathways, and sensory neurons of the CS pathway (*Crow and Tian, 2006*).

In butanone plasticity of *C. elegans,* the odor is analogous to the CS, while the food or food deprivation context could be considered the US. Learning is specific to the conditioned odor, as expected for a classical CS (*Colbert and Bargmann, 1995*). The AIA circuit and *ins-1,* which are required for aversive but not appetitive learning, are needed to combine information about odor and food deprivation (*Figure 7*). We suggest that the AIA circuit senses food deprivation, and retrograde signaling to AWC merges this US information with odor-specific CS information. The AIM neurons that are implicated in long-term appetitive learning might have corresponding roles in encoding food as an appetitive US (*Lakhina et al., 2015*).

## Odor experience shifts the dynamic range of butanone sensitivity in AWC<sup>ON</sup>

Organisms ranging from bacteria to mammals modify the sensitivity and dynamic range of sensory transduction after prolonged stimulation (*Fain et al., 2001*; *Sourjik, 2004*). In AWC<sup>ON</sup>, we observed a strong but transient suppression of calcium responses immediately after odor conditioning (*Chalasani et al., 2010*), followed by a sustained shift in the sensitivity and dynamic range of the odor response. Aversive and appetitive conditioning elicited similar changes favoring higher odor concentrations, suggesting that this change represents sensory adaptation to the average odor intensity in the animal's environment.

The shift in AWC<sup>ON</sup> dynamic range after odor conditioning may prevent saturation at higher odor concentrations, allowing detection of increasing odor concentrations against an odor background. Indeed, the increased butanone attraction after appetitive conditioning is most striking at high odor concentrations (*Torayama et al., 2007*). Naive animals are only weakly attracted to these concentrations, perhaps reflecting the saturation of their AWC<sup>ON</sup> calcium response.

## Sensory neuron synapses as sites of associative plasticity

After aversive conditioning, the AWC<sup>ON</sup> neuron has a diminished ability to drive reversal behavior. We suggest that this results from alterations in synaptic vesicle release by AWC<sup>ON</sup>, which contains both glutamatergic and peptidergic vesicles (*Chalasani et al., 2010*). Candidate mechanisms for plasticity are changes in the composition or level of the neuropeptides produced by AWC, changes in the relative rates of release of the two vesicle classes, or changes in resting potential that affect basal neurotransmitter release. Gene expression changes induced by nuclear EGL-4 or other transcriptional regulators may contribute to these functional effects (*Juang et al., 2013*; *Neal et al., 2015*).

Sensory neuron synapses and related signaling pathways also contribute to gustatory learning in *C. elegans. C. elegans* migrates to salt concentrations that have been paired with food (*Kunitomo et al., 2013*; *Luo et al., 2014*), and learns the association over about 90 min, a time frame similar to that of olfactory learning. Salt preference changes are accompanied by synaptic changes in the ASER gustatory neuron that are induced by INS-1, AGE-1, and an axonal isoform of the DAF-2 insulin receptor (*Kunitomo et al., 2013*; *Oda et al., 2011*; *Ohno et al., 2014*; *Tomioka et al., 2006*). Bidirectional changes in ASER behavioral output below and above the preferred salt concentration are observed in a channelrhodopsin experiment similar to the one reported here for AWC<sup>ON</sup> (*Kunitomo et al., 2013*).

Unlike olfactory learning, however, salt preference learning is accompanied by bidirectional shifts in the salt concentration that activates ASER in calcium imaging experiments (*Kunitomo et al., 2013*; *Luo et al., 2014*). Similarly, *C. elegans* temperature preference learning results in precise bidirectional shifts in the temperature sensitivity of the AFD thermosensory neurons (*Kimura et al., 2004*; *Clark et al., 2006*; *Ramot et al., 2008*; *Yu et al., 2014*). A key difference between olfactory learning and these other forms of plasticity is the nature of sensory preference. In thermotaxis and gustatory plasticity, animals seek out a preferred concentration at the 'setpoint', the temperature or salt condition associated with chronic cultivation (*Hedgecock and Russell, 1975*; *Kunitomo et al., 2013*; *Luo et al., 2014*). Returning to the setpoint is the apparent homeostatic goal of the directed behavior. Olfactory plasticity, by contrast, is relatively insensitive to setpoint. It is linked to the identity of the odor, not the quantity, and the readouts of aversive and appetitive learning are observed across 100-fold concentration changes in the odor source (*Colbert and Bargmann, 1995*; *Torayama et al., 2007*). Although odor history does change sensory representations in AWC<sup>ON</sup>, this is not central to the behavioral preference; it is the context-dependent synaptic changes that appear to be instructive. Individual neurons may tune history-dependent changes in sensory detection and associative changes at sensory synapses in a variety of ways, generating different forms of sensory plasticity to solve different behavioral problems.

# Materials and methods

## Nematode cultures and strains

Animals were grown at 20–22°C on Nematode Growth Medium (NGM) plates seeded with *E. coli* OP50 4–10 days before use. Wild type were Bristol strain N2 hermaphrodites. Standard molecular biology and microinjection methods (*Mello and Fire, 1995*) were used to generate transgenic strains, listed below.

| Strain | Genotype | Figures |
|---|---|---|
| CX15261 | *kyIs617 [gcy-28d::HisCl1::SL2::GFP (5 ng/μl), myo-3::mCherry (5 ng/μl)]* | 1 |
| CX14849 | *kyEx4867 [ins-1(short)::HisCl1::SL2::mCherry (50 ng/μl), unc-122::GFP (10 ng/μl)]* | 1 |
| CX16863 | *kyIs698 [ttx-3intron7::HisCl1::SL2::GFP (30 ng/μl)]* | 1 |
| CX15954 | *kyEx5402 [str-3::HisCl1::SL2::GFP (100 ng/μl)]* | 1 |
| CX15341 | *kyEx5161 [unc-4::HisCl1::SL2::mCherry (50 ng/μl), elt2::mCherry (1 ng/μl)]* | 1 |
| CX16862 | *kyEx4867 [ins-1(short)::HisCl1::SL2::mCherry (50 ng/μl), unc-122::GFP (10 ng/μl)] + kyEx5402 [str-3::HisCl1::SL2::GFP (100 ng/μl)]* | 1 |
| CX17181 | *kyEx6003 [str-3::HisCl::SL2::mCherry (50 ng/μl), myo-3::mCherry (5 ng/μl)] + kyIs698 [ttx-3intron7::HisCl::SL2::GFP (30 ng/ul)]* | 1 |
| CX17183 | *kyEx6005 [unc-4::HisCl::SL2::mCherry (50 ng/ul), myo-3::mCherry (5 ng/ul)] + kyIs698 [ttx-3intron7::HisCl:SL2::GFP (30 ng/ul)]* | 1 |
| CX14908 | *kyEx4924 [inx-1::hisCl1::SL2::GFP (30 ng/μl), myo-3::mCherry (5 ng/μl)]* | 1 |
| CX14909 | *kyEx4925 [ttx-3::hisCl1::SL2::GFP (50 ng/μl), myo-3::mCherry (5 ng/μl)]* | 1 |
| CX16069 | *kyEx5493 [glr-3::HisCl1::SL2::mCherry (50 ng/μl), elt-2::mCherry (1 ng/μl)]* | 1 |
| CX16061 | *kyEx5485 [str-1::HisCl1::SL2::GFP (50 ng/μl)]* | 1 |
| CX15388 | *kyEx5178 [tph-1(short)::HisCl1::SL2::mCherry PCR product (15 ng/μl)]* | 1 |
| CX16040 | *kyEx5464 [tdc-1::HisCl1::SL2::mCherry (50 ng/μl)]* | 1 |
| CX16866 | *kyIs617 [gcy-28d::HisCl1::SL2::GFP (5 ng/μl), myo-3::mCherry (5 ng/μl)] + kyEx5836 [ins-1(long)::dsRNA(HisCl) (25 ng/μl), unc-122::GFP (10 ng/μl)]* | 1 |
| CX16867 | *kyIs617 [gcy-28d::HisCl1::SL2::GFP (5 ng/μl), myo-3::mCherry (5 ng/μl)] + kyEx5837 [unc-4::dsRNA(HisCl) (100 ng/μl), unc-122::GFP (10 ng/μl)]* | 1 |
| CX14599 | *kyEx4747 [gcy-28d::unc-103(gf)::SL2::mCherry (30 ng/μl), elt-2::mCherry (2 ng/μl), pSM (70 ng/μl)]* | 1 |
| CX7155 | *ins-1(nr2091)* | 2 |
| JN1704 | *ins-1(nr2091); peEx1704 [ins-1(short)::ins-1::Venus]* | 2 |
| TJ1052 | *age-1(hx546)* | 2 |
| CX17261 | *age-1(hx546); kyEx6035 [str-2::age-1::SL2::GFP (50 ng/ul)]* | 2 |
| CX16499 | *kyIs678 [odr-3::GFP::egl-4 (5 ng/μl), elt-2::nlsGFP (5 ng/μl), pSM (90 ng/μl)]* | 3 |
| CX16500 | *kyIs678 [odr-3::GFP::egl-4 (5 ng/μl), elt-2::nlsGFP (5 ng/μl), pSM (90 ng/μl)] + kyEx4747 [gcy-28d::unc-103(gf)::SL2::mCherry (30 ng/μl), elt-2::mCherry (2 ng/μl), pSM (70 ng/μl)]* | 3 |
| CX16674 | *age-1(hx546); kyIs678 [odr-3::GFP::egl-4 (5 ng/μl), elt-2::nlsGFP (5 ng/μl), pSM (90 ng/μl)]* | 3 |
| CX13914 | *kyEx4275 [str-2::GCaMP5A (10 ng/μl), unc-122::dsRed (10 ng/μl)]* | 4 |
| CX16213 | *kyEx5527 [str-2::nlsGCaMP6s (30 ng/μl), elt-2::nlsGFP (5 ng/μl)]* | 4, 5 |
| CX16503 | *kyEx4747 [gcy28d::unc-103(gf)::SL2::mCherry (30 ng/μl), elt-2::mCherry (2 ng/μl)] + kyEx5527 [str-2::nlsGCaMP6s (30 ng/μl), elt-2::nlsGFP (5 ng/μl)]* | 5 |
| CX17242 | *age-1(hx546); kyEx5527 [str-2::nlsGCaMP6s (30 ng/μl), elt-2::nlsGFP (5 ng/μl)]* | 5 |
| CX14418 | *kyEx4605 [str-2::ChR2 H134::GFP (50 ng/μl), myo-3:mCherry (10 ng/μl)]* | 6 |
| CX16670 | *kyEx4605 [str-2::ChR2 H134::GFP (50 ng/μl), myo-3:mCherry (10 ng/μl)] + kyEx4747 [gcy-28d::unc-103(gf)::SL2::mCherry (30 ng/μl), elt-2::mCherry (2 ng/μl)]* | 6 |
| CX17243 | *age-1(hx546); kyEx4605 [str-2::ChR2 H134::GFP (50 ng/μl), myo-3:mCherry (10 ng/μl)]* | 6 |
| CX13210 | *kyEx3838 [inx-1:ChR2 H134::GFP (30 ng/μl), unc-122::GFP (20 ng/μl)]* | 6 |

## Chemotaxis assays

Chemotaxis was tested on square plates containing 10 mL of chemotaxis agar (1.6% agar, 5 mM phosphate buffer pH 6.0, 1 mM $CaCl_2$, 1 mM $MgSO_4$), poured the day before the assay. Adult animals were washed three times with S basal buffer and once with chemotaxis buffer (5 mM phosphate buffer pH 6.0, 1 mM $CaCl_2$, 1 mM $MgSO_4$), and 100–200 animals placed at the center of the square plate. Two 1 µl drops of butanone diluted at 1:1000 in ethanol, or ethanol control, were spotted on each side of the plate at the beginning of the assay, with 1 µl 1 M $NaN_3$ at the same spot to immobilize animals that reached the butanone source or ethanol source. After 1–2 hr, plates were moved to 4°C to stop the assay. The assay was quantified by counting animals that had left the origin, on each side of the plate (#Odor, #Control) and in the middle (#Other), and calculating a chemotaxis index as [#Odor - #Control] / [#Odor + #Control + #Other].

The odor concentrations experienced by the animal during chemotaxis are estimated to span nanomolar to micromolar concentrations; the point sources provide 2 µl of 11 mM butanone, which disperses through the 110 ml volume of the chemotaxis plate for an average concentration of 200 nM.

## Aversive olfactory learning

Assay conditions were modified from *Colbert and Bargmann (1995)* and *L'Etoile et al. (2002)*. One-day old adults were washed three times with S basal buffer and placed in 2 ml glass vials (VWR 66009–556) containing 1 ml of S basal buffer with or without butanone diluted to a final concentration of 1 mM. Vials were laid horizontally to maintain aeration during odor conditioning. Previous studies have shown that aversive olfactory learning towards butanone is comparable after conditioning on agar plates or in liquid (*L'Etoile et al., 2002*). After conditioning for 90 min, animals were washed twice with S basal and once with chemotaxis buffer before being tested in butanone chemotaxis assays. Each test population consisted of 50–200 animals, and experiments were repeated a minimum of three times. Plates that had less than 50 animals outside the origin at the end of the assay were excluded from analysis to ensure that the chemotaxis index was an accurate representation of the population.

For histamine experiments, histamine dihydrochloride (Sigma) was added during the conditioning phase at a final concentration of 10 mM as described in *Pokala et al. (2014)*.

## Appetitive olfactory learning

Assays were modified from *Torayama et al. (2007)*. One-day old adults were washed three times with S basal buffer and placed on NGM plates seeded with *E. coli* OP50. 12 µl of undiluted butanone was spotted onto agar plugs on the lid of the plate, and the plate was sealed with parafilm. The effective butanone concentration is approximately the same as the concentration used in aversive conditioning. Mock-conditioned groups were placed on seeded plates without butanone on the lid. After 90 min, animals were washed twice with S basal and once with chemotaxis buffer and assayed for chemotaxis to 2 µl of a 1:10 dilution of butanone (100-fold more than in the aversive olfactory learning assay; the average odor concentration experienced in this chemotaxis assay is 20 µM).

## Cell identification for *gcy-28d* transgene expression, and relevance for interpretation of silencing results

Animals ranging from L2 to adults were mounted on a 2% agar pad containing 5 mM $NaN_3$ under a glass cover slip. The *HisCl1* and *unc-103(gf)* transgenes are coupled to GFP and mCherry sequences, respectively, in bicistronic operons with an SL2 splice leader sequence before the fluorescent protein. Animals were observed under a 60X objective of a Zeiss Axioskop microscope and neurons expressing the fluorescent marker were identified based on known landmarks and morphological characteristics. Animals carrying the *gcy-28d::HisCl1* transgene showed expression in AIA (100%), AVF (~90%), ASI (~65%, weak), and IL2, I1, or M3 pharyngeal neurons (~70% each). Animals carrying the *gcy-28d::unc-103(gf)* transgene showed expression in AIA (~95%), AVF (<5%), ASI (<5%), and IL2, I1, or M3 neurons (~5% each). The *gcy-28d::unc-103* expression pattern suggests that silencing AIA alone can disrupt aversive learning, but this appears inconsistent with the complete set of *HisCl1* data in *Figure 1C*, because aversive learning was normal with two *HisCl1* transgenes that are

expressed in AIA (*ins-1(s), ttx-3(intron7)*). Two potential explanations for this apparent discrepancy are (1) chronic AIA silencing with *unc-103(gf)* disrupts aversive learning, but transient AIA silencing during conditioning with HisCl1 does not unless additional neurons are silenced (2) in addition to AIA, a low level of *unc-103(gf)* in another neuron, below the detection limit of the mCherry reporter, silences that neuron in the *gcy-28d::unc-103(gf)* strain.

## dsRNA-mediated silencing of HisCl expression

Plasmids driving the sense and anti-sense sequences of *HisCl1* under cell-specific promoters were injected at equal concentrations. Double-stranded RNAs in *C. elegans* can act systemically, so control experiments were conducted with similar sense/anti-sense plasmids directed against the GFP marker in the *gcy-28d::HisCl1* strain. These controls showed that the GFP dsRNA expressed from the *unc-4* or *ins-1* promoters silenced expression in AVF or AIA, respectively, without silencing expression in the reciprocal neuron or in other *gcy-28d*-expressing neurons, e.g. in the pharynx. The *ins-1* promoter is also expressed weakly in ASI, and might silence HisCl1 there as well. This possibility was difficult to assess rigorously because the weak *gcy-28d* expression in ASI was already near the detection threshold.

## EGL-4 nuclear translocation assay

A plasmid with the *odr-3* promoter driving GFP::EGL-4 (*Lee et al., 2010*) was injected into wild-type animals at 5 ng/μl. A spontaneous integrant of the resulting extrachromosomal array was back-crossed four times to wild-type N2 animals. This strain had normal chemotaxis to butanone and normal aversive olfactory learning behavior.

One-day old adults were conditioned to butanone in glass vials as described above, washed three times with S basal buffer, and mounted on a 2% agar pad containing 5 mM NaN$_3$ under a glass cover slip. Fluorescence was scored within 20 min to avoid effects of NaN$_3$ on EGL-4 localization. The AWC neuron proximal to the objective was identified using a 40x objective of a Zeiss Axioskop microscope and a Z-stack was taken through the cell using Zeiss Axiovision software for image capture. From each stack, the image containing the central plane of the cell was selected for quantification. Fluorescence values of the cytoplasmic and nuclear regions were quantified using ImageJ software.

25–30 animals per group were imaged during each experiment, and three experiments were conducted for each group. Results were plotted as cumulative distribution plots, and statistical significance measured by the Kolmogorov-Smirnov test.

Butanone conditioning results in EGL-4 translocation only in the AWC$^{ON}$ cell (*Lee et al., 2010*). The strain used here did not have a marker to distinguish AWC$^{ON}$ from AWC$^{OFF}$, because AWC$^{ON}$ – specific promoters were not strong enough to drive EGL-4::GFP reliably, and attempts to include a second AWC$^{ON}$ transgene as a marker with EGL-4::GFP resulted either in interference between the two transgenes, or in defects in butanone chemotaxis or aversive olfactory learning. Therefore, it should be assumed that 50% of the AWC neurons that were imaged were AWC$^{ON}$ and 50% were AWC$^{OFF}$.

## Calcium imaging

For the calcium imaging data in *Figure 4A*, custom-fabricated polydimethylsiloxane devices were used for imaging and stimulus delivery as described in *Chalasani et al. (2007)*. Animals were conditioned outside the device for 90 min and single animals were loaded into the device for calcium imaging. For all other calcium imaging experiments, imaging and stimulus delivery was performed as described in *Larsch et al. (2013)*. Devices with two separate arenas suitable for simultaneous imaging of two conditions or genotypes were flooded with S basal buffer containing 10 mM tetramisole hydrochloride to paralyze the animals, and 6–12 adult animals were loaded into each arena. For aversive conditioning, animals were conditioned in the device for 90 min with 1 mM butanone in S basal buffer, followed by 15 min of S basal buffer without odor. To ensure that butanone absorption by the PDMS device did not affect the results, a smaller number of animals were tested in a control experiment in which animals were conditioned outside the device, washed, and loaded into a fresh imaging device that had not been exposed to odor for calcium imaging. Naive and conditioned animals had calcium responses to 111 nM and 111 μm butanone that were similar to those of animals

conditioned within the device in *Figure 4*. For appetitive conditioning, animals were conditioned outside the device as in the behavioral assay, because introducing *E. coli* into the microfluidic devices contaminates the arena. 2–4 experiments were conducted for each group.

Odor stimuli were delivered using a three-way valve to switch between buffer and odor flow into the arena, and a Hamilton valve to switch between different odor concentrations. The stimulation protocol consisted of three 30-second alternations between buffer and odor at each concentration, followed by one minute of buffer, and then the sequence was repeated at a 10-fold higher odor concentration for a total of six butanone concentrations ranging from 11 nM to 1 mM. Calcium responses to odors were monitored in animals expressing GCaMP5A (*Figure 4A*; *Akerboom et al., 2012*), or nuclear-localized GCaMP6s (*Figures 4B–D*, *5*; *Chen et al., 2013*) under the *str-2* promoter, which is selectively expressed in AWC[ON] neurons. Fluorescence was monitored on a Zeiss AxioObserverA1 microscope with a 2.5X objective, and Metamorph software was used for synchronized image capture with pulsed illumination.

## Imaging data analysis

Fluorescence was measured using a custom ImageJ script as in *Larsch et al. (2013)*.

Custom Matlab scripts were written to plot the fluorescence response over time as well as measure response magnitude and recovery time (see *Source code 1*). Integrated fluorescence values were used. $\Delta F/F_o$ was generated by dividing fluorescence values by the baseline fluorescence, which was determined to be the fluorescence at t = 1700 s (*Figure 4A*). $\Delta F/F_{max}$ was generated by normalizing each trace on a 0–1 scale, where 0 and 1 are defined as the mean of the lowest and highest 5% of pixel intensities, respectively (all other Figures). AWC response magnitude was measured as the decrease in normalized fluorescence following the first odor addition in the series, which was calculated by subtracting the average fluorescence in the last 10 s during the odor presentation from the average fluorescence from the 2 s preceding the odor addition. Half-time of recovery was defined as the time for fluorescence to recover to 50% of the peak magnitude after the last odor removal in the series, and was only calculated when there was a response to odor (defined as a reduction in fluorescence of magnitude 0.075 or greater). Averages in *Figure 4D,5C,F and I* were only calculated when at least half of the individuals responded to odor. Naive animals did not recover fully from the last 1 mM pulse within the five minutes of recording (*Figure 4B*, *5A,C*), so for this concentration the half-time of recovery is a lower bound.

## Channelrhodopsin-induced behavior

L4-stage animals expressing the H134R variant of Channelrhodopsin 2 (*Lin et al., 2009*) under the *str-2* (AWC[ON]) or *inx-1* (AIB) promoter were incubated overnight on plates seeded with *E. coli* OP50 and 50 μM retinal. The following day, adult animals were conditioned with odor and with or without food as described above. After conditioning, animals were washed twice with S basal and once with NGM buffer (25 mM phosphate buffer pH 6.0, 1 mM $CaCl_2$, 1 mM $MgSO_4$, 52 mM NaCl) and transferred to a 6 cm NGM plate with a square filter ring soaked in 20 mM $CuCl_2$ to prevent animals from crawling out of the field of view. Animals received 20-second pulses of blue light (455 nm, 25 μW/mm$^2$) every two minutes, repeated ten times, and were video-recorded; pulses six through ten were analyzed. A Pixelink camera and Streampix software were used to generate recordings. Movies were analyzed using custom Matlab scripts that tracked animal locomotion and identified reversals (*Gordus et al., 2015*). Custom Matlab scripts were used to bin and plot the frequency of events over time (see *Source code 2*). Each group was tested on 7–14 experimental trials, with 15–25 animals in each trial.

## Acknowledgements

We thank Oliver Hobert, Yuichi Iino and Douglas Kim for reagents and Xin Jin and Donovan Ventimiglia for discussion and comments on the manuscript. CEC was supported by a fellowship from the Department of Defense. CIB is an investigator of the Howard Hughes Medical Institute (HHMI). This work was supported by HHMI.

## Additional information

### Funding

| Funder | Grant reference number | Author |
|---|---|---|
| Howard Hughes Medical Institute | | Cornelia I Bargmann |
| U.S. Department of Defense | National Defense Science and Engineering Graduate Fellowship | Christine E Cho |
| National Institute on Deafness and Other Communication Disorders | R01DC005991 | Noelle D L'Etoile |

The funders had no role in study design, data collection and interpretation, or the decision to submit the work for publication.

### Author contributions

CEC, Designed, performed and interpreted experiments, Co-wrote the paper; CB, NDL, Contributed reagents and participated in the design of the EGL-4 nuclear enrichment studies, Aided in drafting of the manuscript; CIB, Designed and interpreted experiments, Co-wrote the paper

### Author ORCIDs

Cornelia I Bargmann, http://orcid.org/0000-0002-8484-0618

## Additional files

### Supplementary files

• Source code 1. Calcium Imaging Suite: Image J and Matlab scripts for tracking and analysis of calcium imaging assays in *Figures 4* and *5*.

• Source code 2. Optogenetics Assay Suite: Matlab scripts for tracking and analysis of channelrhodopsin-induced turning assays in *Figure 6*.

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
