## [Decision Letter]

Thank you for submitting your article "Different forms of plasticity encode sensory history and context in *Caenorhabditis elegans* olfactory learning" for consideration by *eLife*. Your article has been reviewed by two peer reviewers, and the evaluation has been overseen by Mani Ramaswami as Reviewing Editor and K VijayRaghavan as the Senior Editor.

The reviewers have discussed the reviews with one another and the Reviewing Editor has drafted this decision to help you prepare a revised submission.

Cho et al. probe the mechanisms that underlie two forms of olfactory learning induced when the odorant butanone is presented to *C. elegans* either without or with food. A series of chemogenetic inactivation and optogenetic activation experiments, augmented by calcium imaging from key sensory neurons and interneurons in the olfactory circuit, shows that negative conditioning occurs downstream of primary chemosensory neurons and requires activation of a specific set of interneurons. Sites where associative context and sensory input converge are proposed.

The quality of the experiments is excellent and the writing is clear. However, in its present form the paper lacks a clear narrative that conveys any broad implications necessary for a general interest journal like *eLife*. In current form, it is not made clear why the findings should be of interest to those outside the field of *C. elegans* neurobiology.

While appreciating that the authors may have their own ideas here, the reviewers, in an effort to be constructive, suggested three possible ways in which the work could be elaborated or presented to achieve wider interest: (1) To focus on the question of how two mechanisms of learning are differentially achieved in one circuit (comparisons with the recent *Drosophila* literature could be useful). (2) To convincingly propose potentially universal molecular or mechanistic insight (e.g. on retrograde signaling). (3) To present the work in context of some general problem: e.g. of plasticity of sensory-motor circuits involved in guiding food-seeking behavior. The Introduction and Discussion as well as the presentation of the Results will need to be revised accordingly.

Major points:

1) Is "adaptation" the best term to describe *C. elegans* olfactory learning associated with food-deprivation for 80 minutes? It would help increase the broad impact of the paper by using terminology more widely accepted in the literature on learning and memory. While there is a long history in *C. elegans* of using "adaptation" to refer to the reductions in chemotaxis behavior observed in response to prolonged exposure to attractants, it seems that now may be a time to reconsider this term. If food deprivation has strong negative consequences (if the worms feel strong hunger and fat stores are depleted as suggested by some previous papers) then perhaps this should be termed aversive conditioning as the worms on blank plates are actually learning a conditioned association between odor and starvation? This should be clarified early, as it would help justify and explain subsequent parallels being drawn to better-established forms of aversive conditioning. It also seems reasonable that associations formed between butanone and a rich source of food should be referred to as "appetitive conditioning." The suggested change in terminology would also make the paper easier to understand, by avoiding the many different meanings of "adaptation."

2) The EGL-4 story could be better integrated into the paper. To better communicate the summary model (Figure 7), one of the key premises of the work should be made clear in the Introduction: that EGL-4 translocation is both necessary and sufficient for persistent "behavioral adaptation." A related limitation that the authors should discuss is that the evidence for the sufficiency part of the premise may be weak. In the cited work, Lee et al. 2010 report that constitutive nuclear localization of EGL-4 "renders the animals unresponsive to AWC-sensed odors." The concern is that constitutive expression may have larger and/or additional/different effects than acute translocation to the nucleus during an actual adaptation experiment. With respect to the model, it is also recommended that the authors provide one or more plausible scenarios (ideally with precedents in the literature) for how nuclear EGL-4 might reduce the effectiveness of AWC activation.

3) Since silencing AIA or ASI neuron separately did not reduce adaptation the authors believe that AIA plus ASI neurons contribute to adaptation. Therefore, they should assess the effects of silencing these two neurons simultaneously using the *ins-1::HisCl1*, which should drive HisCl1 expression in both AIA and ASI

4) The claim/premise that the AIA circuit is butanone selective is overstated.

Other points:

5) Overall impact is lessened by the rather vague definition of the AIA et al. circuit. This vagueness is not the authors' fault, merely a result of how the experiments turned out, but it's a problem nonetheless.

6) It would help to have the term "context" formally defined at first use. To some, it carries the strong connotation of "surrounding external cues," whereas it is used here (as of course in many other cases) to denote internal state (satiety/starvation).

7) Most readers outside the *C. elegans* field will not be able to make the connection between changes in the downstream effectiveness in AWC and changes in the chemotaxis scores by which adaptation and enhancement are quantified.

8) It is often unclear when "adaptation" means: sensory adaptation (reduction in gain of sensory transduction) versus behavioral change (reduction in the strength of chemotaxis). It would help greatly to use different terms to refer to cell biological and behavioral changes.

9) The term "starvation" connotes an internal metabolic state. Is "food deprivation" both a more neutral and correct term? If "starvation" is used, as indeed may be more appropriate, then this should be justified on the basis of existing experimental data.

10) In the last paragraph of the subsection “AWC output reflects bidirectional odor preference”. The claim that the AIB::ChR2 experiment localizes the site of adaptation to somewhere between AWC depolarization and AIB depolarization doesn't seem to take into account the parallel pathways from AIY, another follower of AWC, to the reverse command neurons such as AVA. From the experiment we learn that loci postsynaptic to AIB are unlikely but there are many possible sites postsynaptic to AIY.

11) Figure 1. How is the two-way ANOVA applied? The text should clarify which main effect is significant in each case or in general. Or if interactions were the significant thing, then this should be indicated.

12) In the fourth paragraph of the subsection “AIA and other neurons are required for behavioral adaptation”. The authors say that restoring AVF activity in the *gcy-28d::HisCl* transgene does not rescue (restore?) adaptation. This claim is presumably made on the basis of the similarity between the Naive + histamine and Adapted + histamine groups. But a closer look also suggests the chemotaxis index of the Naive + histamine is pretty close to the adapted level. So another valid interpretation is that the Adapted + histamine group IS adapted, such that restoring AVF active actually rescue "adaptation." The authors should please clarify this point and discuss it more completely.

---

## [Author Response]

*The quality of the experiments is excellent and the writing is clear. However, in its present form the paper lacks a clear narrative that conveys any broad implications necessary for a general interest journal like eLife. In current form, it is not made clear why the findings should be of interest to those outside the field of C. elegans neurobiology.*

While appreciating that the authors may have their own ideas here, the reviewers, in an effort to be constructive, suggested three possible ways in which the work could be elaborated or presented to achieve wider interest: (1) To focus on the question of how two mechanisms of learning are differentially achieved in one circuit (comparisons with the recent Drosophila literature could be useful). (2) To convincingly propose potentially universal molecular or mechanistic insight (e.g. on retrograde signaling). (3) To present the work in context of some general problem: e.g. of plasticity of sensory-motor circuits involved in guiding food-seeking behavior. The Introduction and Discussion as well as the presentation of the Results will need to be revised accordingly.

To bring forward a clear narrative that conveys broad implications to a general-interest reader, we have made the following changes to the manuscript.

First, we emphasized the idea that we are observing both non-associative and associative learning that occur at the same time in the same circuit, beginning with the abstract. This highlights the “two mechanisms of learning achieved in one circuit” as well as addressing the work “in the context of some general problem” as suggested.

Second, we addressed molecular mechanisms more deeply by adding experiments examining the insulin/PI3 kinase pathway in aversive learning. Previous work from our group and others implicated retrograde signaling from insulin to sensory neurons in aversive learning. We now show that retrograde insulin/PI3 kinase signaling from AIA to AWC affects aversive learning to butanone (Figure 2); that this signaling pathway is required for the nuclear translocation of the EGL-4 cGMP-dependent kinase in AWC during conditioning (Figure 3); that the insulin/PI3K signaling pathway is not required for non-associative sensory adaptation (Figure 5); and that the insulin/PI3K signaling pathway is required for the acute behavioral changes relevant to associative aversive learning (Figure 6). Thus insulin/PI3K signaling, and its unanticipated effect on the EGL-4 kinase, mark the convergence of CS (odor) and US (food deprivation) in AWC.

*Major points:*

1) Is "adaptation" the best term to describe C. elegans olfactory learning associated with food-deprivation for 80 minutes? It would help increase the broad impact of the paper by using terminology more widely accepted in the literature on learning and memory. While there is a long history in C. elegans of using "adaptation" to refer to the reductions in chemotaxis behavior observed in response to prolonged exposure to attractants, it seems that now may be a time to reconsider this term. If food deprivation has strong negative consequences (if the worms feel strong hunger and fat stores are depleted as suggested by some previous papers) then perhaps this should be termed aversive conditioning as the worms on blank plates are actually learning a conditioned association between odor and starvation? This should be clarified early, as it would help justify and explain subsequent parallels being drawn to better-established forms of aversive conditioning. It also seems reasonable that associations formed between butanone and a rich source of food should be referred to as "appetitive conditioning." The suggested change in terminology would also make the paper easier to understand, by avoiding the many different meanings of "adaptation."

We have followed the reviewers’ advice and rewritten the paper using the phrases “aversive olfactory learning” and “appetitive olfactory learning” instead of the historical terms “adaptation” and “enhancement”. The word “adaptation” is now used solely for non-associative sensory adaptation.

2) The EGL-4 story could be better integrated into the paper. To better communicate the summary model (Figure 7), one of the key premises of the work should be made clear in the Introduction: that EGL-4 translocation is both necessary and sufficient for persistent "behavioral adaptation." A related limitation that the authors should discuss is that the evidence for the sufficiency part of the premise may be weak. In the cited work, Lee et al. 2010 report that constitutive nuclear localization of EGL-4 "renders the animals unresponsive to AWC-sensed odors." The concern is that constitutive expression may have larger and/or additional/different effects than acute translocation to the nucleus during an actual adaptation experiment. With respect to the model, it is also recommended that the authors provide one or more plausible scenarios (ideally with precedents in the literature) for how nuclear EGL-4 might reduce the effectiveness of AWC activation.

We have integrated EGL-4, whose nuclear enrichment is required for long-term aversive learning, into the paper more fully. We show that insulin/PI3K signaling regulates EGL-4 localization in AWC neurons. This result ties together insulin and EGL-4, two genetic requirements for aversive learning whose relationship was previously unknown. In the Discussion we describe the known ability of nuclear EGL-4 to change gene expression, and suggest that aversive learning changes the expression of molecules required for glutamatergic or peptidergic neurotransmission by AWC.

3) Since silencing AIA or ASI neuron separately did not reduce adaptation the authors believe that AIA plus ASI neurons contribute to adaptation. Therefore, they should assess the effects of silencing these two neurons simultaneously using the ins-1::HisCl1, which should drive HisCl1 expression in both AIA and ASI

We have added results to Figure 1 that show that simultaneous acute silencing of AIA and ASI is not sufficient to block aversive learning. Combining transgenes for simultaneous silencing of AIA and AVF caused aversive learning defects even without histamine, making the result uninterpretable.

4) The claim/premise that the AIA circuit is butanone selective is overstated.

We removed the point about butanone specificity.

*Other points:*

5) Overall impact is lessened by the rather vague definition of the AIA et al. circuit. This vagueness is not the authors' fault, merely a result of how the experiments turned out, but it's a problem nonetheless.

Vague definition of AIA circuit: We try to describe it in precise language, but acknowledge that some neurons remain to be identified.

6) It would help to have the term "context" formally defined at first use. To some, it carries the strong connotation of "surrounding external cues," whereas it is used here (as of course in many other cases) to denote internal state (satiety/starvation).

“Context” is now defined as “food” or “food deprivation” wherever it appears.

7) Most readers outside the C. elegans field will not be able to make the connection between changes in the downstream effectiveness in AWC and changes in the chemotaxis scores by which adaptation and enhancement are quantified.

The relationship between AWC-induced reversals and chemotaxis is described explicitly in the revised text.

8) It is often unclear when "adaptation" means: sensory adaptation (reduction in gain of sensory transduction) versus behavioral change (reduction in the strength of chemotaxis). It would help greatly to use different terms to refer to cell biological and behavioral changes.

We eliminated “adaptation” except with respect to non-associative sensory adaptation.

*9) The term "starvation" connotes an internal metabolic state. Is "food deprivation" both a more neutral and correct term? If "starvation" is used, as indeed may be more appropriate, then this should be justified on the basis of existing experimental data.*

We replaced “starvation” with “food deprivation” as suggested.

10) In the last paragraph of the subsection “AWC output reflects bidirectional odor preference”. The claim that the AIB::ChR2 experiment localizes the site of adaptation to somewhere between AWC depolarization and AIB depolarization doesn't seem to take into account the parallel pathways from AIY, another follower of AWC, to the reverse command neurons such as AVA. From the experiment we learn that loci postsynaptic to AIB are unlikely but there are many possible sites postsynaptic to AIY.

We changed the interpretation of the AIB:channelrhodopsin experiment to say that the effects of aversive conditioning are mediated “upstream of or parallel to AIB.”

11) Figure 1. How is the two-way ANOVA applied? The text should clarify which main effect is significant in each case or in general. Or if interactions were the significant thing, then this should be indicated.

Comparisons for ANOVA are included in the figure legends.

12) In the fourth paragraph of the subsection “AIA and other neurons are required for behavioral adaptation”. The authors say that restoring AVF activity in the gcy-28d::HisCl transgene does not rescue (restore?) adaptation. This claim is presumably made on the basis of the similarity between the Naive + histamine and Adapted + histamine groups. But a closer look also suggests the chemotaxis index of the Naive + histamine is pretty close to the adapted level. So another valid interpretation is that the Adapted + histamine group IS adapted, such that restoring AVF active actually rescue "adaptation." The authors should please clarify this point and discuss it more completely.

We now mention the possibility that AVF may affect chemotaxis.